# Research on the mechanical properties of EPS lightweight soil mixed with slag

Lifang Mei[1,2], Dali Xiang[1]*, Yiwen Huang[1]

1 School of Civil Engineering, Architectural and Environment, Hubei University of Technology, Wuhan, China
2 Hubei Ecological Road Engineering Technology Research Center, Wuhan, China

* 102110966@hbut.edu.cn

**Data Availability Statement:** All relevant data are within the manuscript and its Supporting Information files.

**Funding:** The author(s) received no specific funding for this work.

## Abstract

Expanded polystyrene (EPS) bead lightweight soil composites are a new type of artificial geotechnical material with low density and high strength characteristics that can be widely used in engineering projects. To promote the wide application of EPS bead lightweight soil in engineering, when slag is used to replace part of the cement as a binding agent, it can better improve the effect of soil and reduce engineering costs. The mechanical properties of EPS lightweight soil mixed with slag were analyzed by conducting an unconfined compressive strength (UCS) test and triaxial test on lightweight soil with different EPS bead contents and slag contents. The particle sizes of the EPS beads are 1~3 mm, the EPS contents are 1%, 2%, 3%, and 4%, and the slag-cement composite binding agents are 10%, 15%, 20% and 25%. The results show that the UCS decreases significantly with increasing EPS bead content at different EPS bead contents and slag contents; the UCS of the specimen with 30% slag content is the largest; and the UCS of lightweight soil without slag is comparable to that of lightweight soil with a slag content of approximately 60%. The peak stress in triaxial increases with increasing confining pressure, and the modulus of deformation decreases linearly with increasing EPS bead content. the slag-cement composite binding agent has a significantly better reinforcing effect than single mixed cement. The stress–strain curves of EPS lightweight soil mixed with slag exhibits hardening and softening characteristics. EPS bead content and slag content determine the stress–strain characteristics of the EPS lightweight soil mixed with slag. The macromechanical properties based on the microscopic mechanism of the EPS lightweight soil mixed with slag shows that different slag contents affect the failure pattern of EPS lightweight soil mixed with slag. The research results can provide a reference for engineering design and application.

## 1. Introduction

With the rapid advancement of modern industry, the utilization of polymer materials is increasingly widespread, and EPS beads are a new type of geotechnical material produced in this context. The EPS molecular structure is stable and does not release harmful substances or cause chemical reactions in the soil. EPS lightweight soil is a kind of geosynthetic material that is mainly composed of original soil, cement, EPS and water mixing and is characterized by

**Competing interests:** The authors have declared that no competing interests exist.

light weight, high strength and durability. Due to the mixing of EPS, the density of lightweight soil is greatly reduced, generally up to 1.2 g/cm$^3$ or less, and compared with the general soil, the density of EPS lightweight soil is small [1, 2], the EPS bead content is the main factor determining the size of the density of the lightweight soil, and the cement content has a smaller effect. The mixing of the binding agent gives the lightweight soil a certain strength and deformation resistance, and the content of EPS beads and binding agent is the main factor affecting the compressive strength of this material. Other factors have less influence on the strength, and the mechanical properties mainly depend on the proportion of the lightweight soil mixture [3–5]. Therefore, the strength of the lightweight soil is adjustable, and it can be adjusted according to the needs of the project. To improve the applicability of lightweight soil, to facilitate the use of local materials, to reduce transportation and construction costs, the waste soil excavated from the pit of the nearby construction site was selected as the original soil, mixed with EPS beads and binding agent, and made into EPS lightweight soil. Due to its light weight and high strength, it plays a vital role in weak foundations, preventing highway settlement, stabilizing slopes, backfilling bridge abutments and underground cavities [6–9], and practical application in developed countries such as Japan [10].

Worldwide, scholars have conducted some studies on EPS lightweight soil and achieved certain research results. Edinçliler et al. [11] studied the effects of EPS bead size, EPS bead volume content and confining pressure on the stress–strain characteristics of EPS lightweight soils by means of triaxial compression tests. Salahaldeen et al. [12] evaluated the properties of hardened concrete, such as the compressive strength and density of the mixed specimens. The results showed that the addition of EPS significantly reduced the mechanical properties of concrete; the range of compressive strength at 28 d of curing was 13.6–1.96 MPa; the addition of EPS beads reduced the weight of the concrete. Yong et al. [13] used EPS beads and silica fume (SF) to replace sand and cement in brick production. The results showed that the EPS beads reduced the compressive strength and density, while the SF strengthened the mixture, thus compensating for the loss of properties caused by the EPS beads.

Studies on the mass incorporation of slag into EPS lightweight soils are rare, and the most commonly used binding agent is cement, but the production of cement causes serious environmental problems and consumes a large amount of energy in the production process [14]. Slag, as an industrial byproduct from a wide range of sources, is relatively inexpensive, and the use of slag instead of cement as a binding agent can both reduce the cost of foundation treatment and minimize the impact on the environment [15, 16]. YANG et al. [17] analyzed the composition and properties of slag and studied the application of slag in the fields of building materials, sewage treatment, agriculture and resource utilization. The results show that making full use of the secondary resources of slag can improve the protection of the environment, realize the harmless treatment of slag, and further improve the residual value of slag. Slag, as an industrial waste material, has been widely used in concrete projects, and by replacing part of the cement in the concrete mix with slag, the construction cost of materials and related energy costs have been reduced, and the mechanical properties and durability of concrete have been significantly improved [18–20]. Zhao et al. [21] and others used cement and slag to cure coastal saline soils, and the reinforcing effect was better than that of cement alone. Liang [22] and Nu et al. [23] conducted indoor tests to show that replacing part of the cement with slag can substantially increase the compressive strength of cement soils, and it was found that increasing the amount of slag admixture can improve the strength of cement soils. Lal et al. [24] proposed mixing bottom ash, EPS beads, and binder into specific geomaterials, and the results showed that for each mix ratio, the compressive strength increased with curing time, and the prepared geomaterials were relatively light in weight and could be used as an alternative to traditional filler materials. Deshmukh et al. [25] suggested that mixing EPS beads with

**Table 1. Physical-mechanical parameters of original soil.**

| Natural density $\rho$/g·cm$^{-3}$ | Void ratio e | Water content $w$(%) | Plastic limit $W_P$(%) | Plasticity index $I_p$ | Liquid limit $W_L$(%) | Specific gravity Gs |
|---|---|---|---|---|---|---|
| 1.917 | 0.544 | 21.79 | 20.29 | 18.5 | 38.79 | 2.64 |

industrial waste could be used to build flexible road surfaces. Lightweight soil is widely used as a new lightweight fill material due to its light weight and high strength. Previous studies have focused on the behavior of various soils mixed with EPS beads and cement. Slag is a common building material used in engineering and construction, and mixing slag in EPS lightweight soil can reduce the amount of cement, lower filling costs, and improve the strength of lightweight soil, achieving economic and environmental protection. Currently, there is a lack of research on EPS lightweight soil mixed with slag, and slag will be developed into a high value-added filling material to achieve large-scale utilization of slag, accelerate the transformation and upgrading of waste utilization, and produce a new type of lightweight filling material. Therefore, it is necessary to carry out research on the mechanical properties of EPS lightweight soil mixed with slag.

In this study, the clay in Wuhan, Hubei Province is used as the original soil, EPS beads as the lightweight material, slag and cement are mixed, and different slag mixing amounts are prepared to mix the binding agent to form an EPS lightweight soil mixed with slag and to research the mechanical properties of EPS lightweight soil mixed with slag with different material ratios. The mechanical properties and failure morphology of EPS lightweight soil mixed with slag were investigated by unconfined compressive strength (UCS) tests and consolidated undrained triaxial (CU) tests. Scanning electron microscopy was used to study the microstructure to explore the macromechanical mechanism, and the research results can provide a reference for engineering design and application.

## 2. Materials and methods

### 2.1 Material selection

The original soil used in the test is silty clay, taken from a project pit in Wuhan City. The depth of the soil is approximately 5 m below the ground, and its basic physical parameters are shown in Table 1. The lightweight materials are EPS spherical beads produced by the Plastic Foam Factory, diameter 1~3 mm, pure particle density of 0.024 g/cm$^3$, and packing density of 0.016 g/cm$^3$. The binding agent used 42.5-grade ordinary Portland cement and s95 granulated slag, and the chemical composition of the materials is shown in Table 2. Before the test, the retrieved soil specimens were dried at a temperature range of 105~110°C for a period of no less than 8 h. The dry soil was crushed and passed through a sieve with a 1 mm aperture, and then the sieved dry soil was put into a plastic bag and sealed.

### 2.2 Specimen preparation

The ratio of specimens will be based on a mass ratio standard (ratio of admixture of other materials to the mass of dry soil). The EPS beads were mixed at 1%, 2%, 3%, and 4%, and the

**Table 2. Chemical compositions of cement and Slag $w$/%.**

| Composition | CaO | $Fe_2O_3$ | $SiO_2$ | $Al_2O_3$ | MgO | $SO_3$ | Loss |
|---|---|---|---|---|---|---|---|
| Cement | 61.23 | 3.28 | 22.56 | 6.25 | 2.28 | 1.67 | 1.02 |
| Slag | 34 | 1.03 | 34.5 | 17.7 | 6.01 | 1.64 | 0.84 |

slag-cement composite binding agent was mixed at 10%, 15%, 20% and 25%. The numerical value of slag as a percentage of binding agent was 0%, 10%, 30%, 50%, and 70%, and water was tap water.

Dry soil, slag and cement were added in a certain proportion in a sample bucket and mixed thoroughly according to 50% water content, mixed with water, stirred into a homogeneous mixed slurry, and then added to EPS beads and stirred for 10 minutes to form a homogeneous EPS lightweight soil mixed with slag (for the convenience of the narrative, this soil can be referred to as lightweight soil). The lightweight soil was loaded into a molding die that was 80 mm high and 39.1 mm in diameter, compacted in 5 layers, and turned with a soil scraper between each layer and to a depth of not less than 1 cm until a dense specimen was formed. Finally, the prepared specimen, together with the specimen maker, was placed into a standard curing box at a curing temperature of $20 \pm 2°C$ and a relative humidity greater than 95% after 24 h of curing demolding, and continued curing to the design age of 28 d. Each group of tests was used to produce three parallel specimens, and the average was taken as the final result.

In this paper, the above lightweight soil specimens were tested for UCS concerning the "Standard for Geotechnical Test Methods (GB/T-50123-2019)" [26], and the instrument used was the WDW-10E microcomputer-controlled electronic universal testing machine, with a strain rate of 1 mm/min. The CU triaxial compression test was carried out with a strain-controlled triaxial compression apparatus, which was taken to have a strain rate of 0.08 mm/min. The confining pressures of the test were taken as 50, 100, 200, and 300 kPa. During microscopic testing, the test specimens were cut into thin slices for drying after reaching the curing period. After that, the specimens were broken into small squares, 3 samples were taken from each specimen, and the specimens were coated. The scanning electron microscope technique was used to analyze the microstructure of the lightweight soil.

## 3. Results and discussion

### 3.1 Effect of EPS content on the density of lightweight soil

EPS beads are lightweight materials with good performance, light weight, seismic properties, low cost, etc. EPS added to the soil body can significantly reduce the soil body's own load to achieve the role of reducing soil pressure and load. In the test to consider the effect of EPS beads and binding agent on the density, the EPS bead content was 1%, 2%, 3%, and 4%, and the binding agent content was 10%, 15%, 20%, and 25%. The ratio of slag to cement was taken as 1:1, a total of 16 groups of specimens, and the average density of the lightweight soil after 28 d of curing was analyzed in the test. Fig 1 shows the relationship between different EPS beads, binding agent mixing amounts, and lightweight soil. From Fig 1, it can be seen that with the changes in binding agent and EPS content, the density of lightweight soil is the minimum of 0.841 g/cm$^3$, and the density is the maximum of 1.49 g/cm$^3$. When the EPS bead content is unchanged, with the increase in binding agent content, the trend in density increases slightly. In lightweight soil with a 5% difference in binding agent mixing, the difference in density is only 0.096–0.112 g/cm$^3$, which shows that the effect of the binding agent content on the density is very small. When the amount of binding agent is unchanged, the density decreases significantly with increasing EPS content, and the density decreases by at least 20% for every 1% increase in the content of EPS beads. However, as the EPS content increases from 1% to 2%, the density decrease is the largest. Taking the binding agent addition of 10% as an example, when the EPS content increases from 1% to 4%, the density decreases from 1.436 g/cm$^3$ to 0.841 g/cm$^3$, which is a reduction of 41.5%, indicating that the density of the lightweight soil decreases significantly. This law shows that it is feasible to realize lightweight soil by adding EPS beads; thus, it can be considered that the density of lightweight soil is mainly affected by

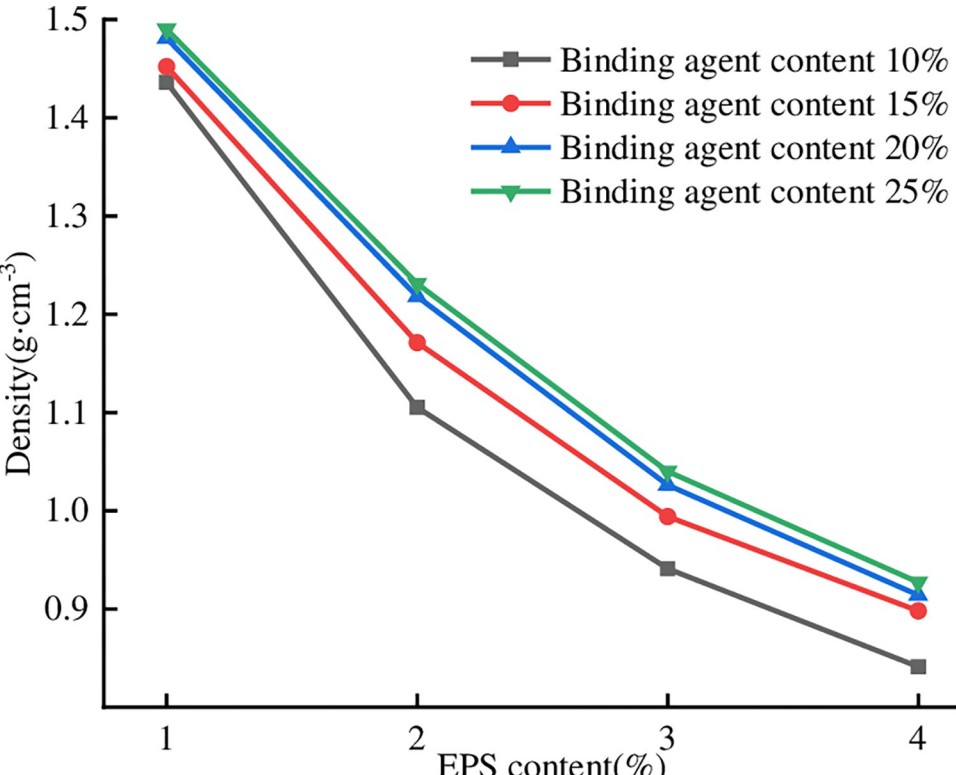

**Fig 1. Density versus EPS content.**

the content of EPS beads, and the content of the binding agent is not the main factor influencing density.

The amount of EPS beads cannot be increased indefinitely in the mixture. The density of lightweight soil cannot be reduced indefinitely, and the excess EPS beads lead to a reduction in the bonding effect of the binding agent. It is difficult to make a specimen, and lightweight soil loses the value of practical application.

## 3.2 Stress-strain properties of clays

Fig 2 shows the stress–strain curve for the silty clay. The UCS test of the clay was added for comparison with the lightweight soil that follows. Three specimens were taken for parallel tests, and the average value was taken as the final value. The clay reaches the peak stress, and then the stress decays rapidly, indicating strain softening. The maximum stress is 214 kPa.

## 3.3 Effect of EPS content on unconfined compressive strength

The addition of lightweight materials and binding agents results in lightweight soils that are different from natural soils but instead resemble porous cemented soils. The truly novel aspect of this geomaterial is the inclusion of lightweight materials that create a large number of cavity structures within the soil mix, thereby greatly reducing its weight but still providing a certain level of strength. In Fig 3, the content of the binding agent is 10%, 15%, 20%, and 25%, the content of EPS beads is 1%, 2%, 3%, and 4%, the ratio of slag to cement is 1:1, and the UCS curves of a total of 16 groups of specimens are taken. The UCS of different EPS contents has a similar trend, with increasing EPS content, the strength decreases. Especially when the EPS content is

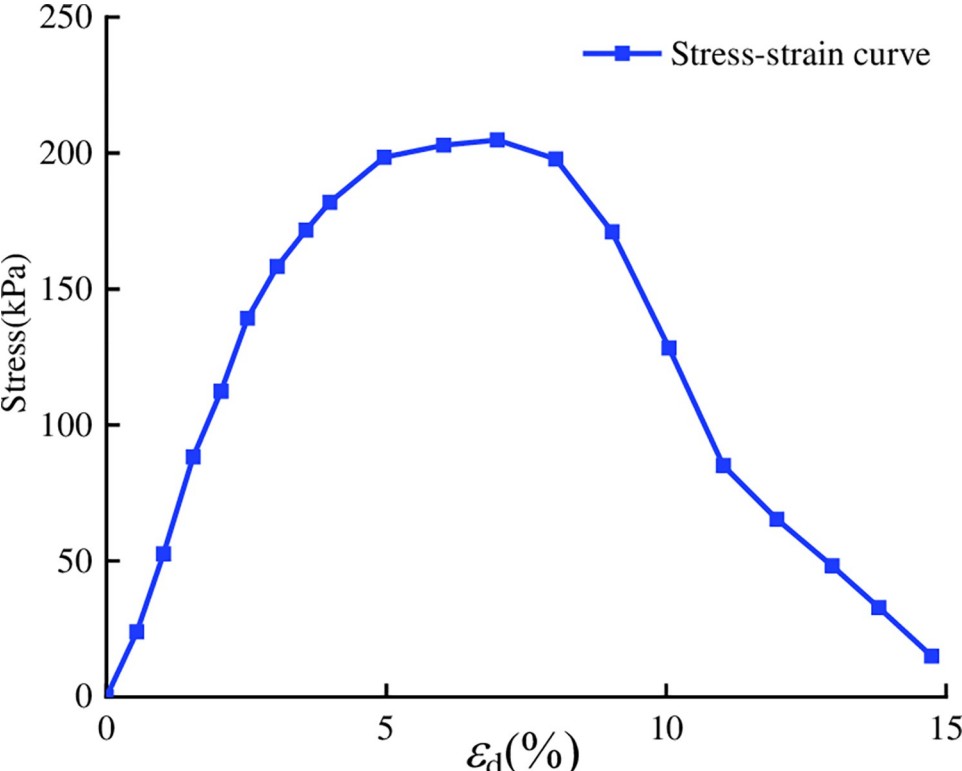

**Fig 2. Silty clay stress–strain curve.**

less than 2%, the strength decreases more, for example, when the content of the binding agent is 25% when the EPS content is increased from 1% to 2%, the strength decreases from 1464.1 kPa to 600.5 kPa, which is a decrease of 58.9%; when the EPS content is more than 3%, the strength also decreases significantly, but the trend of decrease is slower; when the EPS content is 4%, the strength is 205.7 kPa, which is a decrease of 85.9%. Therefore, the weight cannot be reduced by increasing the amount of EPS without limitation, which will have a greater impact on the compressive strength.

Fitting the data points in Fig 3 reveals that the exponential curve fits well with a correlation coefficient $R^2$ greater than 0.96, which agrees with the findings of Mei [27] on lightweight soils, where the relationship between the UCS and the EPS admixture is:

$$q_{u} = q_0 e^{-ta_e} \tag{1}$$

In Formula (1), $q_{u}$ is the UCS; $a_e$ is the EPS content; and $q_0$ and $t$ are the fitting parameters for the relationship between strength and EPS content, respectively, and are related to the binding agent content. Both $q_0$ and $t$ increase with increasing binding agent content.

### 3.4 Effect of binding agent content on unconfined compressive strength

The EPS content was selected as 1%, and the relationship curves between different binding agent contents and UCS were plotted. The slag content of the specimens in Fig 4 was 0%, 10%, 30%, 50%, and 70%, and the binding agent content was 10%, 15%, 20%, and 25%. When the slag content was constant, the UCS at different binding agent contents had the same trend, and the strength gradually increased with increasing binding agent content. The UCS at 25%

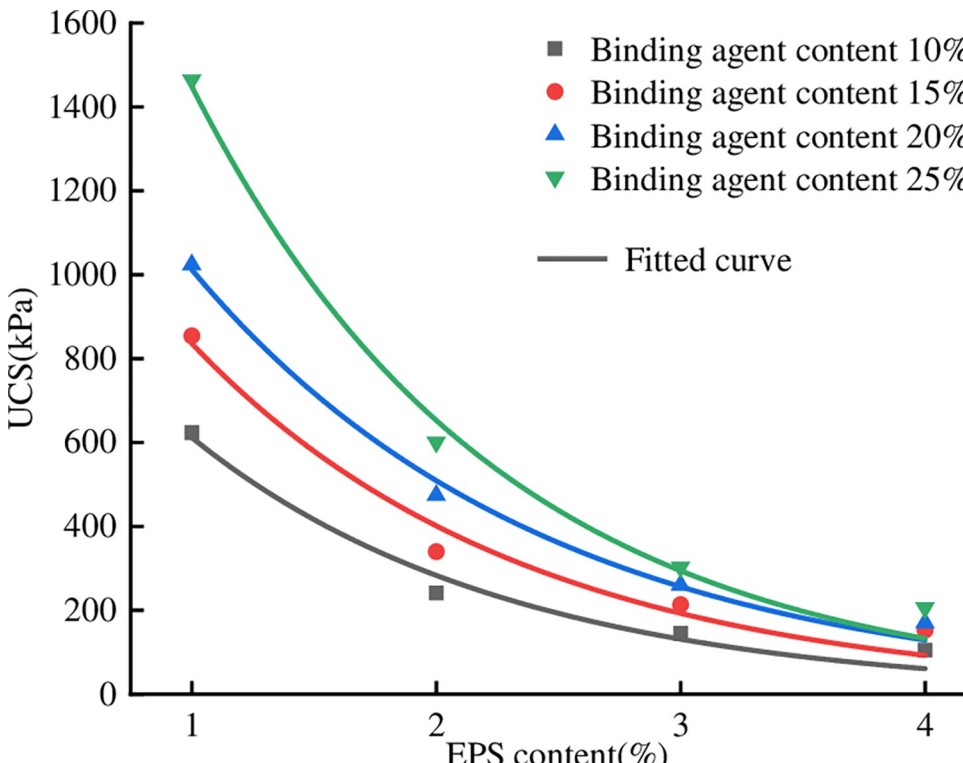

**Fig 3. Relationship between UCS and EPS content.**

binding agent content was 1.96, 2.14, 2.15, 2.35, and 2.56 times the UCS of the same group of specimens with 10% binding agent content. The slag content only affects the slope and intercept of the linear relationship. Fitting the data points in Fig 4, the correlation coefficient $R^2$ is greater than 0.93, which is a good fit. This is consistent with the findings of HOU [28] on lightweight soils. The relationship between the UCS and the content of the binding agent is:

$$q_u = ka_c + b \qquad (2)$$

In Formula (2), $a_c$ is the binding agent content, and k and b are both fitting parameters for the relationship between strength and binding agent content, which is related to slag content.

### 3.5 Effect of slag content on unconfined compressive strength

The binding agent content was selected to be 20% unchanged, and the slag content was 0%, 10%, 30%, 50%, and 70%. Fig 5 shows that with increasing slag content, the UCS curves of the lightweight soils increased and then decreased, which indicates that the slag content is not the best. At the same EPS content, the UCS of the specimens reached the maximum value at 30% slag content; when the EPS content was 1%, 2%, 3%, and 4%, compared with the specimens without slag content (binding agent was cement), the UCS of the specimens at 30% slag content increased by 534.9 kPa, 400.7 kPa, 360.4 kPa and 212.4 kPa, respectively. The strength of the specimens without slag content is comparable to that of the specimens with approximately 60% slag content. This shows that it is feasible to utilize slag to partially replace cement as a binding agent, but the amount of replacement should be within a suitable range. Compared with the lightweight soil without slag, the compressive strength of the lightweight soil with a certain percentage of slag is greater than that without slag at the same EPS content.

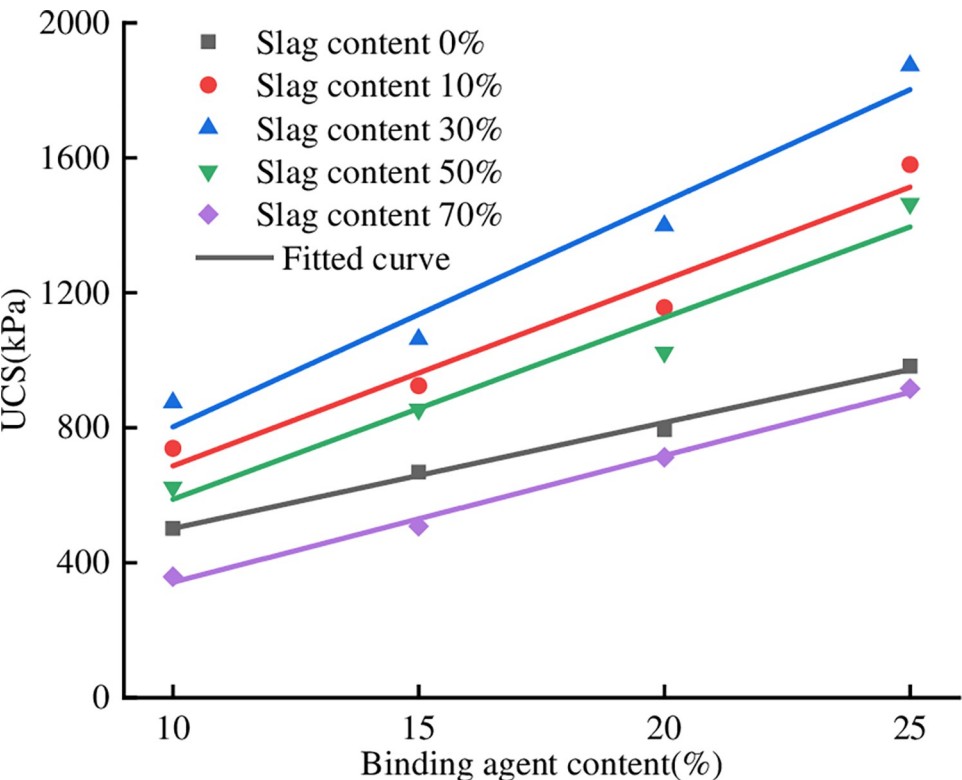

**Fig 4. Relationship between UCS and binding agent content.**

### 3.6 Triaxial test analysis

The cured specimens were vacuum saturated for 2 h, and after 24 h of immersion in water, they were not fully saturated, the saturation degree was 60% to 85%, and the specimens continued to be saturated with counterpressure in the triaxial instrument. The specific test program is shown in Table 3.

The stress–strain curve obtained from the test is shown in Fig 6, where $\sigma_1$ is the large principal stress, $\sigma_3$ is the small principal stress, $\varepsilon_d$ is the axial strain, and $a_g$ is the slag content. From Fig 6, it can be seen that in the case of the same amount of each content, the peak principal stress difference ($\sigma_1$-$\sigma_3$) increases with the increase of the confining pressure, the stress–strain curve under different confining pressure has nonlinear characteristics, with the increase of strain, the slope of the curve decreases, and the overall performance of strain hardening characteristics; when the confining pressure is the same, the strength decreases with an increase in EPS bead content, and the curve morphology is transformed from the softening type to the hardening type. In the case of the same amount of EPS content, the strength is affected by slag content, and 30% of slag content has the highest strength, which indicates that the ratio has a very important influence on the stress–strain relationship. Fig 6(E) in the test with 3% EPS content and 50% slag content, the curves are of the softening type when the confining pressure is 50 kPa and 100 kPa, indicating that the specimens with stronger cemented structure exhibit softening and hardening characteristics when the confining pressure is at the relatively low level. The lower level exhibits softening characteristics, which are is not an accidental phenomenon. Similar test phenomena can be observed from the research data of the literature [29].

From the test results, the initial stage of the stress–strain relationship curve of lightweight soil is a straight line, and elastic deformation occurs. After reaching the yield stress, the

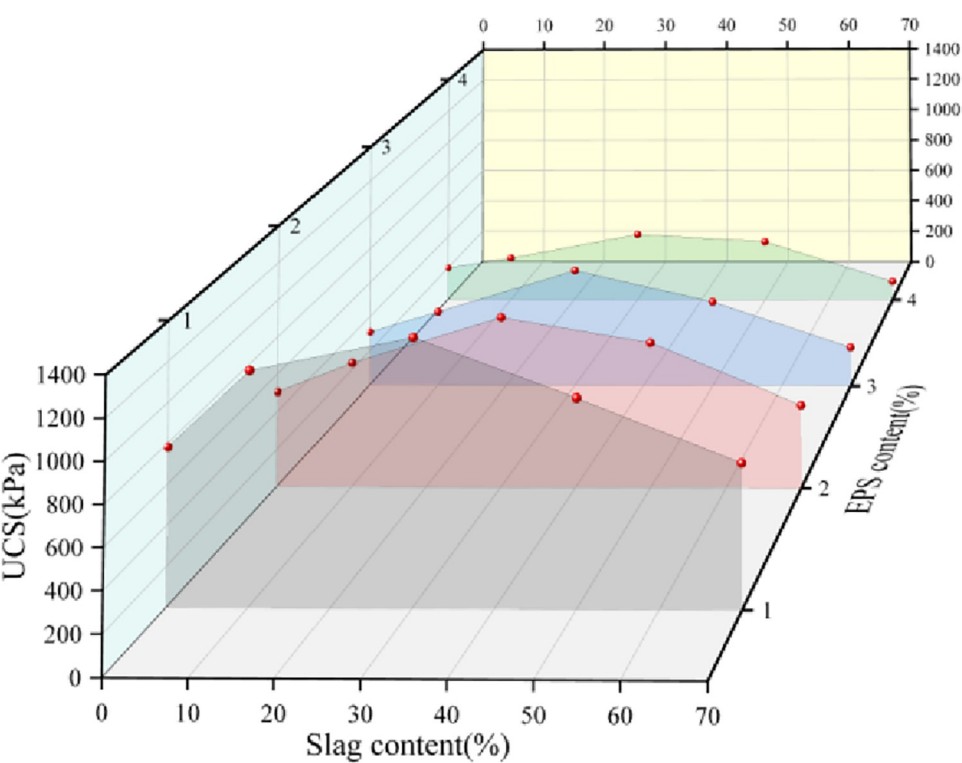

**Fig 5. Relationship between UCS and slag content.**

specimen undergoes elastic–plastic deformation, and the stress–strain relationship shows non-linear characteristics. The research results of Pei [30] and others showed that the stress–strain relationship of more structural clay was strain-hardening when the confining pressure was higher than the structural yield stress and strain-softening when the confining pressure was lower than the structural yield stress. Hou [31] used silt to prepare lightweight soil and conducted a CU test to obtain the result that when the confining pressure is less than the structural yield stress of the lightweight soil specimen, the strain softening phenomenon is more obvious; when the confining pressure is greater than the structural yield stress of the specimen, the strain hardening phenomenon is more obvious. Although a large number of EPS beads were mixed in, the particles of lightweight soil were combined very tightly due to the net-like cementation structure formed by the binding effect of the binding agent and entered the recompression and compacting stage under the high confining pressure, showing a tendency of increasing strength. In contrast, under the limiting effect of low confining pressure, the stress of lightweight soil increases with strain to a certain degree and then slowly decreases to a certain residual strength, and the stress–strain curve exhibits a hump curve, i.e., strain soften-ing type. While Dong et al. [32] used Nanjing silty powdery clay as the original soil to make lightweight soil specimens (under the condition of comparable ratios), the CU test curve pat-terns were all of the strain-hardening type, which indicates that the transformation of the

**Table 3. Triaxial compression test (CU) scheme of lightweight soil.**

| Scheme | EPS content (%) | Slag content (%) | Water content (%) | Binding agent content (%) | Confining pressure (kPa) |
|--------|-----------------|------------------|-------------------|---------------------------|--------------------------|
| 1      | 2, 3, 4         | 30               | 50                | 20                        | 50, 100, 200, 300        |
| 2      | 3               | 10, 30, 50       | 50                | 20                        | 50, 100, 200, 300        |

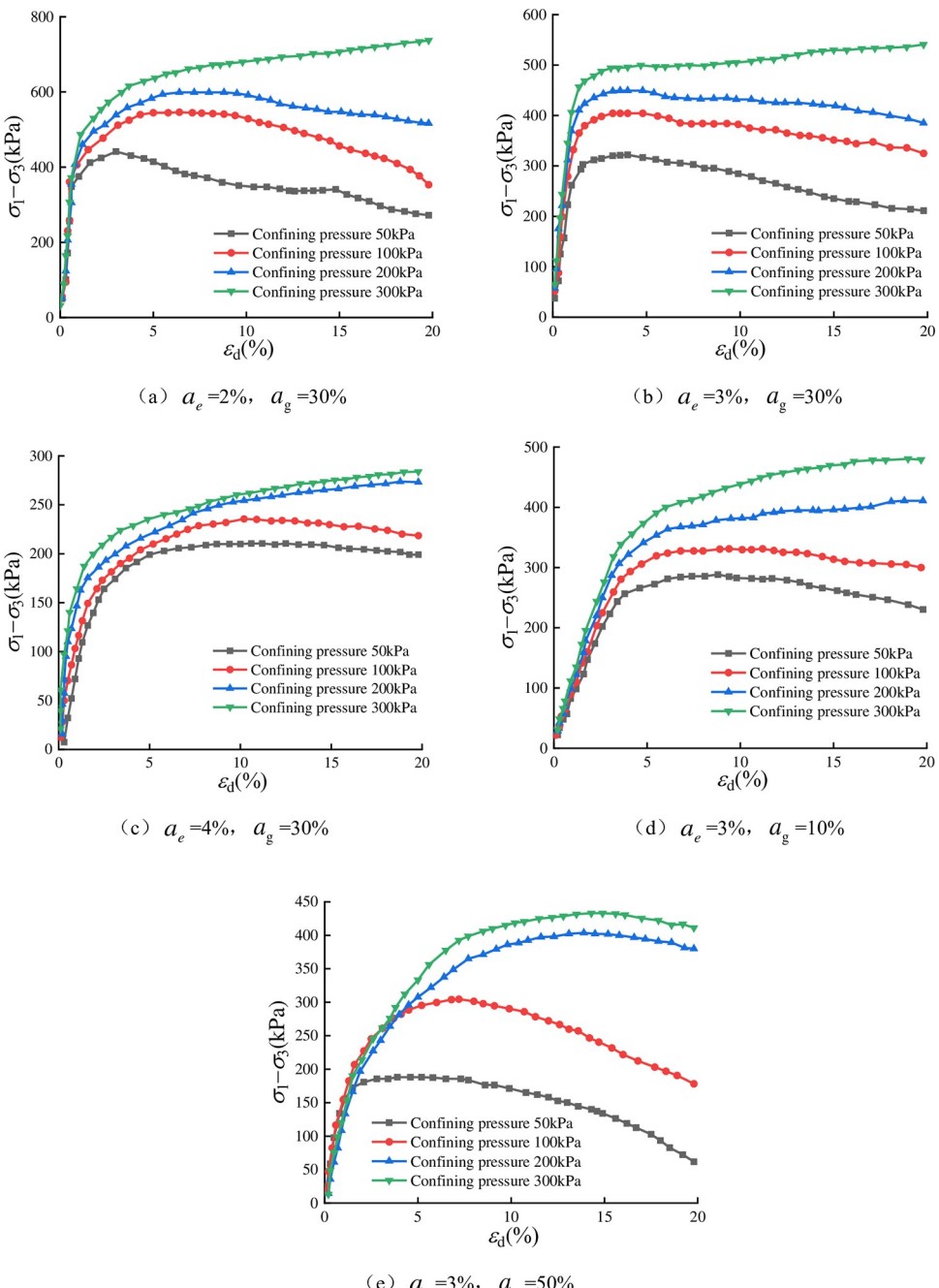

**Fig 6. Stress-strain relationship curves of lightweight soils with different ratios.**

stress–strain relationship of the lightweight soils and the type and nature of the original soils also have a certain relationship.

## 3.7 Analysis of factors affecting the deformation modulus of lightweight soils

Due to the compression and shear expansion of the soil, the modulus of deformation $E$ is not a constant but a parameter that varies with the stress level. The average deformation modulus

$E_{50}$, which is the slope of the cut line of the stress–strain curve from 0 to 50% of the compressive strength, is usually used in engineering and is calculated as follows:

$$E = \frac{1/2\sigma_{max}}{\varepsilon_{1/2}} \tag{3}$$

In Formula (3), $\sigma_{max}$ is the compressive strength and $\varepsilon_{1/2}$ is the strain corresponding to 50% of the compressive strength.

The average deformation modulus of the soil body is related to the EPS bead content, binding agent content, and confining pressure. To simplify the problem, other factors were fixed, linear regression analysis was performed on a single factor, the correlation coefficient $R^2 = 0.958$, and the deformation modulus of specimens with different EPS admixtures and different enclosing pressures showed good normalization. As shown in Fig 7. The deformation modulus of lightweight soil decreased with increasing EPS content, the discrete points were fitted, and the results showed that there was a linear relationship between the two, with the following relationship equation.

$$E = ka_e + c \tag{4}$$

In Formula (4), $k = -0.302$, $c = 2.225$, $a_e$ is the EPS bead content, and the deformation modulus is important for the study of soil subsidence and foundation settlement.

### 3.8 Unconfined compression failure patterns

The failure of the lightweight soil specimen reflects the structural nature of the soil body to a certain extent. For the analysis of the results of the UCS test, it was concluded that the

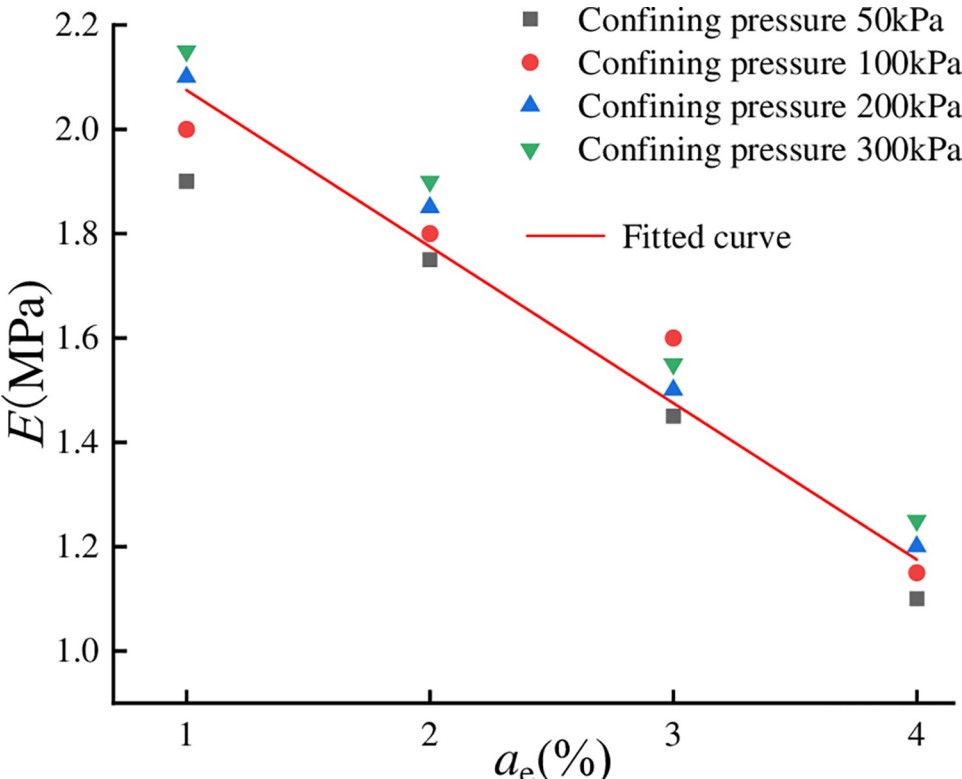

**Fig 7. Relationship between deformation modulus and EPS content.**

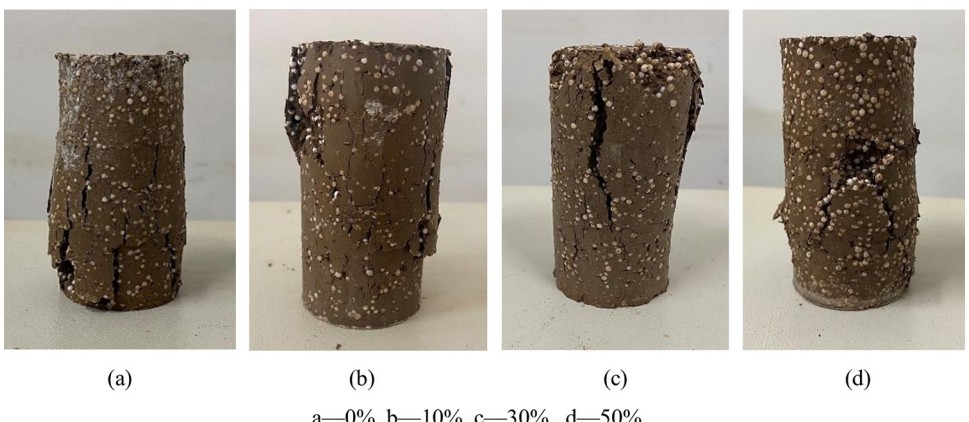

(a) (b) (c) (d)

a—0%, b—10%, c—30% , d—50%

**Fig 8. Damage patterns of specimens with different slag contents.** a–0%, b–10%, c–30%, d–50%.

specimens had two failure modes, cracking failure and shear failure, and the specimens had obvious rupture surfaces. The representative specimens were selected based on a binding agent content of 20% and an EPS content of 2%. Fig 8(A) The specimen has an obvious rupture surface, the test has dropped blocks or loose broken phenomenon, and the specimen is cracking failure. Fig 8(B) shows that the oblique surface of the specimen experiences shear failure, the rupture surface inclination is slightly slower, and multiple cracks fail. Fig 8(C) shows that the specimen strength is higher, the pressure produces a penetrating crack, the shear band is obvious, and the rupture surface inclination (and the large principal stress surface angle) is very steep, showing shear failure. Fig 8(A) and 8(D) of the failure pattern are the same, and the specimen failure appears bulging and belongs to the cracking failure.

Through observation and analysis, under the action of an external load, the holes inside the specimen and the interface of EPS beads are the first to experience stress concentration, the shear band cuts through these parts, and the EPS beads themselves do not undergo shear failure, but from the rupture surface macroscopically, it can be observed that the EPS beads undergo a certain degree of deformation. The specimens showed different failure patterns depending on the content of slag and cement. The lightweight soil without slag was easily loose and broken, and after adding an appropriate amount of slag, the structural and mechanical properties of the lightweight soil were enhanced.

### 3.9 Characterization of the microstructure of lightweight soils

The physic mechanical properties of soil are essentially controlled by its internal structure, and any of its complex physic mechanical traits are the comprehensive manifestation of microstructural properties and their changes. The mechanism of macroscopic mechanical behavior of soil can be understood through the in-depth study of the microstructural characteristics of the soil body [33, 34]. To further study the slag-cement-cured lightweight soil, scanning electron microscopy (SEM) observation of lightweight soil with different contents was carried out to analyze the microstructure of lightweight soil with different compositions of binding agent action through electron micrographs. The specimens with 1% EPS beads and 20% binding agent content were selected for observation.

Fig 9 is a microscopic picture of the combination of mixed soil and EPS beads. The surface of the EPS bead is wrapped with a large number of crystals, which connects the EPS beads and soil particles tightly. The EPS beads have a hollow honeycomb structure, with a large number

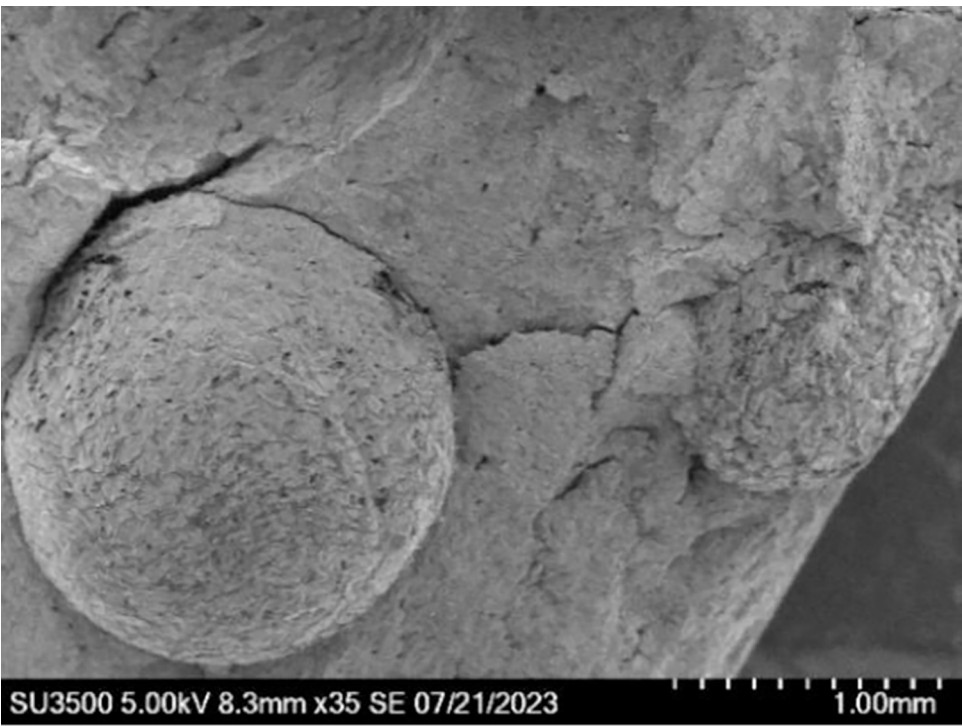

**Fig 9. EPS microscopic morphology.**

of pores, and therefore are light in mass, and the joint action with the binding agent endows the soil with lightweight and high-strength characteristics.

As seen in Fig 10A, the interparticle pores of the slag-free specimen are greater, and the cement in the soil body undergoes a hydration reaction, resulting in reticulated cementitious products. Fig 10B shows the slag into the specimen interparticle pores filled by smaller slag particles. Lightweight soil contains a large number of crystals, the crystals are more net-like distribution in the soil, and the structural units have a more agglomerated structure, the formation of a certain skeleton filling effect, the soil structure of the skeleton of the degree of densification, can be seen binding agent plays a role in the soil as a colloid, which changes the type of association between soil particles from contact bond to cementation bond. The cementation bond strengthens the bond strength between soil particles, so that the structural strength of the soil body shows an increasing trend. As shown in Fig 10C, with increasing slag content, the skeleton filling effect is more significant, the pore space between soil particles is smaller, the soil body is more compact, the generation of hydrated crystals is also gradually increased, and the hydrated crystals are wrapped around the soil particles to bond so that the macroscopic mechanical properties are greatly improved. Fig 10D shows that as the slag content increases, the generation of hydrated crystals in the soil increases significantly, but the pore space and lightweight soil structure become loose instead, and the structural strength becomes weaker.

## 4. Conclusions

1. The density of EPS lightweight soil mixed with slag is mainly affected by the EPS content and decreases nonlinearly with increasing EPS content, and the decrease gradually slows.

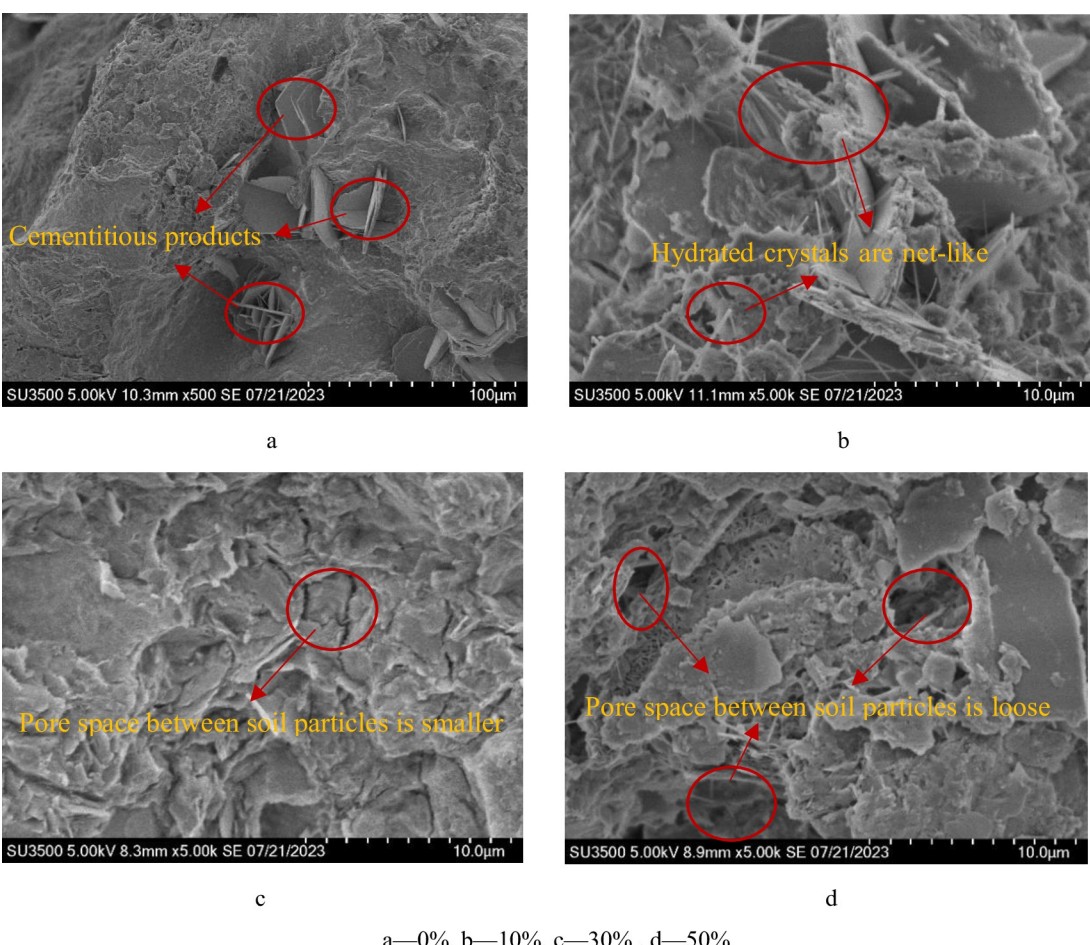

a—0%, b—10%, c—30% , d—50%

**Fig 10. SEM photography of the specimens with different proportions of slag as a binding agent.** a–0%, b–10%, c–30%, d–50%.

2. The unconfined compressive strength decreases nonlinearly with increasing EPS content, increases linearly with increasing binding agent content, and increases with increasing slag content, with the highest strength at 30% slag content.

3. The stress–strain curves of EPS lightweight soil mixed with slag with different ratios show hardening and softening characteristics under different confining pressures. The triaxial stress–strain characteristics are essentially determined by the combined effect of the material ratio, which determines the strength of the cemented structure, and the confining pressure, which determines the stress state.

4. The reticular cementation structure formed by the internal hydration reaction of EPS lightweight soil mixed with slag is the main source of its strength, which directly affects the mechanical properties of lightweight soil. Compared with lightweight soil without slag, lightweight soil with the addition of an appropriate amount of slag has enhanced soil body integrity and increased strength.

In this study, the effect of soil properties, EPS bead size, and type of binding agent on the mechanical properties of EPS lightweight soil may be of particular significance, and the mechanical properties of this composite will be systematically investigated in future studies.

## Supporting information

**S1 Table.**
(XLSX)

**S1 Data.**
(RAR)

## Author Contributions

**Formal analysis:** Yiwen Huang.

**Writing – original draft:** Dali Xiang.

**Writing – review & editing:** Lifang Mei.

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
