## [Decision Letter · Decision Letter 0]

7 Nov 2023

PONE-D-23-31063Research on Mechanical
Properties of Slag EPS Mixed Lightweight SoilPLOS
ONE

Dear Dr. Xiang,

Thank you for submitting your manuscript to PLOS ONE. After careful consideration, we
feel that it has merit but does not fully meet PLOS ONE’s publication criteria as it
currently stands. Therefore, we invite you to submit a revised version of the
manuscript that addresses the points raised during the review process.

Please submit your revised manuscript by Dec 22 2023 11:59PM. If you will need more
time than this to complete your revisions, please reply to this message or contact
the journal office at plosone@plos.org. When
you're ready to submit your revision, log on to https://www.editorialmanager.com/pone/ and select the 'Submissions
Needing Revision' folder to locate your manuscript file.

Please include the following items when submitting your revised
manuscript:A rebuttal letter that responds to each point raised by the academic
editor and reviewer(s). You should upload this letter as a separate file
labeled 'Response to Reviewers'.A marked-up copy of your manuscript that highlights changes made to the
original version. You should upload this as a separate file labeled
'Revised Manuscript with Track Changes'.An unmarked version of your revised paper without tracked changes. You
should upload this as a separate file labeled 'Manuscript'.If you would like to make changes to your financial disclosure,
please include your updated statement in your cover letter. Guidelines for
resubmitting your figure files are available below the reviewer comments at the end
of this letter.

We look forward to receiving your revised manuscript.

Kind regards,

Abdullah Ekinci, PhD

Academic Editor

PLOS ONE

Journal Requirements:

Reviewers' comments:

Reviewer's Responses to Questions

**Comments to the Author**

1. Is the manuscript technically sound, and do the data support the conclusions?

Reviewer #1: Yes

Reviewer #2: Partly

2. Has the statistical analysis been performed
appropriately and rigorously? 

Reviewer #1: No

Reviewer #2: No

3. Have the authors made all data underlying the
findings in their manuscript fully available?

Reviewer #1: Yes

Reviewer #2: Yes

4. Is the manuscript presented in an intelligible
fashion and written in standard English?

Reviewer #1: No

Reviewer #2: No

5. Review Comments to the Author

Reviewer #1: * English needs to be improved

* EPS definition need to be given in abstract

* particle size and their % need to be given in abstract

*Rewrite paragraph on line 36

* Rewrite section 2.1 to be more understandable

* line 141. unlimited increase? what do you mean?

* Table 3 scheme need to be capital letter and why 20% curing agent is decided to be
worked on?

* it may be better to have 3D- with 2 y axes to see effect of agent and EPS graphs on
results

* line 258. cut line? rewrite the sentence

* line 303 physic mechanical?

* line 310 why those % values are chosen for SEM

* in conclusion part, after giving results, conclusion need to be given to understand
why this study has been conducted

Reviewer #2: 1-Language and Terminology needs rigorous improvement. Some of the
sentences appear to be unclear or ambiguous. The language used must be precise and
that technical terms must be appropriately defined and explained. A formal proof
read is necessary for the whole manuscript. Some of the ambiguous parts are
highlighted but the rest also need checking. A format check is also strongly
advised.

2-The novelty and the research question of the work is unclear. Why and where do we
need light soil and in which structures it could be applied? What is the
application? What is the contribution of the research and its novelty in the field
of geotechnical engineering. What specific gap does this study fill, and how does it
advance the current understanding of the subject?

3- The methodology needs improvement. Which standards are used? How the EPS is
prepared? What are the properties of EPS? What type of clay is used?

4- The clay here seems to CL (?), it is not clear why it needed the improvement at
the first place.

5- The addition of EPS seems to decrease the strength of samples, how the usage of
EPS is justified then?

6- There isn't a comparison with the original soil with no additive, it's likely that
the soil with no admixture would exhibit better properties. A comparison with the
plain soil must be added.

7- what is it meant by curing agent? Slag and cement are most commonly used as
binding agents.

8-EPS could may cause deformation and settlement within time and loading, how the
authors justify the usage of it?

9- Is there any environmental effects of using EPS? slag within the soil? It most be
considered.

6. PLOS authors have the option to publish the peer
review history of their article (what does this mean?). If published, this will
include your full peer review and any attached files.

If you choose “no”, your identity will remain anonymous but your review may still be
made public.

**Do you want your identity to be public for this peer review?** For
information about this choice, including consent withdrawal, please see our
Privacy Policy.

Reviewer #1: No

Reviewer #2: No

---

## [Author Response · Author response to Decision Letter 0]

1 Dec 2023

Dear Editors and Reviewers:

Thank you for your letter and for the reviewers’ comments concerning our manuscript
entitled "Research on Mechanical Properties of Slag EPS Mixed Lightweight Soil"(ID:
PONE-D-23-31063). Those comments are all valuable and very helpful for revising and
improving our manuscript, as well as the important guiding significance to our
research. We have studied comments and have made revision carefully. Revised
portions are marked in red. We sincerely hope this manuscript will be finally
acceptable to be published. Thank you very much for all your help.

Please find the following Response to the comments of reviewers:

Reviewer #1: 

* English needs to be improved

Response: Thanks for your kind reminder. We have carefully checked the spelling,
language, grammar, and punctuation, corrected some errors, and polished the whole
manuscript. We hope our manuscript can meet the journal’s standard.

* EPS definition need to be given in abstract 

Response: Thank you for your careful review and valuable comments. we have added the
EPS definition. “Expanded polystyrene (EPS)” has been given in abstract.

* particle size and their % need to be given in abstract 

Response: Thanks so much for your suggestion. we have added the information about
particle size and their % in abstract. The particle sizes of EPS beads are 1~3mm，the
EPS content are 1%, 2%, 3%, 4%, the slag-cement composite binding agent are 10%,
15%, 20% and 25%, respectively. on line 14-16.

*Rewrite paragraph on line 36

Response: Thanks so much for your comment. We have re-written paragraph on line 36,
and this section is much better. 

* Rewrite section 2.1 to be more understandable

Response: Thank you for your careful review and valuable comments. We have re-written
section 2.1, and this section is more understandable. 

* line 141. unlimited increase? what do you mean?

Response: Thanks so much for your comment. With the increase of EPS beads, the
density of lightweight soil will decrease, too much EPS is difficult to make a
specimen, the strength of lightweight soil will also decrease, The EPS content
cannot be unlimited increased, a suitable ratio is needed to meet the needs of the
project.

* Table 3 scheme need to be capital letter and why 20% curing agent is decided to be
worked on?

Response: Thank you for your careful review and valuable comments. We have modified
the letter in Table 3, the word "curing agent" has been replaced with "binding
agent". the 20 % binding agent is a typical representative.

* it may be better to have 3D- with 2 y axes to see effect of agent and EPS graphs on
results

Response: Thanks so much for your suggestion. based on your suggestion, we have added
3D- with 2 y axes to see effect of agent and EPS graphs, this section is more
specific and clearer. on line 236

* line 258. cut line? rewrite the sentence

Response: Thanks so much for your comment. We are very sorry for our incorrect
writing, the "cut line" means Secant, we have corrected the words.

* line 303 physic mechanical?

Response: Thank you for your careful review and valuable comments. English expression
error, replace " physic mechanical " with " physical and mechanical properties
".

* line 310 why those % values are chosen for SEM

Response: Thanks so much for your comment. In microscopic test, the main observation
is the state of EPS, its % value does not affect its state, a representative % value
can be chosen, 20% binding agent is a suitable % value for microscopic test.

* in conclusion part, after giving results, conclusion need to be given to understand
why this study has been conducted 

Response: Thank you for your careful review and valuable comments. The significance
and value of conducting this study is given after the conclusion. on line
388-390.

Thank you again for your positive and constructive comments and suggestions on our
manuscript. We tried our best to improve the manuscript and made some revisions in
the manuscript. We hope you can accept our revised manuscript.

Reviewer #2: 

1-Language and Terminology needs rigorous improvement. Some of the sentences appear
to be unclear or ambiguous. The language used must be precise and that technical
terms must be appropriately defined and explained. A formal proof read is necessary
for the whole manuscript. Some of the ambiguous parts are highlighted but the rest
also need checking. A format check is also strongly advised. 

Response: Thanks for your kind reminder. We have carefully checked the spelling,
language, grammar, and punctuation, corrected some errors, and polished the whole
manuscript. Technical terms are given appropriate definitions and explanations. We
hope our manuscript can meet the journal’s standard.

2-The novelty and the research question of the work is unclear. Why and where do we
need light soil and in which structures it could be applied? What is the
application? What is the contribution of the research and its novelty in the field
of geotechnical engineering. What specific gap does this study fill, and how does it
advance the current understanding of the subject?

Response: Thank you for your careful review and valuable comments. on line 84-93, I
have modified and supplemented the above formulation. Lightweight soil is widely
used as a new lightweight fill material due to its light weight and high strength.
Previous studies focus on the behavior of various soils mixed with EPS beads and
cement. Slag is a common building material used in engineering and construction,
mixing slag in EPS lightweight soil can reduce the amount of cement, lower filling
costs, and improve the strength of lightweight soil, achieving economic and
environmental protection. Currently there is a lack of research on EPS lightweight
soil mixed with slag, the slag will be developed into a high value-added filling
materials to achieve large-scale utilization of slag, accelerate the transformation
and upgrading of waste utilization, and produce a new type of lightweight filling
material. Therefore, it is necessary to carry out research on the mechanical
properties of EPS lightweight soil mixed with slag.

3- The methodology needs improvement. Which standards are used? How the EPS is
prepared? What are the properties of EPS? What type of clay is used?

Response: Thanks so much for your comment. on line 105, 107-108, 135-136, 147-149, I
have modified and supplemented the above formulation and other relevant places. The
standard is " Standard for Geotechnical Test Methods (GB/T-50123-2019)". The
lightweight materials are EPS spherical beads, produced by Plastic Foam Factory,
diameter 1~3mm, pure particle density of 0.024g/cm3, and packing density of
0.016g/cm3, with light weight, seismic, low cost, etc. EPS added to the soil body
can significantly reduce the soil body's own load, to achieve the role of reducing
soil pressure and load. The original soil used in the test is silty clay.

4- The clay here seems to CL (?), it is not clear why it needed the improvement at
the first place.

Response: Thanks so much for your comment. The original soil in this case is silty
clay. The subject of our research is lightweight soil, a new type of artificial fill
material. on line 37-38, 40-41, 45-51, I have modified and supplemented the above
formulation and other relevant places. With the increase of EPS beads, the density
of lightweight soil is greatly reduced, generally up to 1.2g/cm3 or less. The mixing
of binding agent makes the lightweight soil have certain strength and deformation
resistance. In order to improve the applicability of lightweight soil, to facilitate
the use of local materials, to reduce transportation and construction costs, the
waste soil excavated from the pit of the nearby construction site was selected as
the original soil, mixed with EPS beads and binding agent, and made into EPS
lightweight soil. Due to its light weight and high strength, it plays a vital role
in weak foundations, preventing highway settlement, stabilizing slopes, backfilling
bridge abutments and underground cavities. 

5- The addition of EPS seems to decrease the strength of samples, how the usage of
EPS is justified then?

Response: Thank you for your careful review and valuable comments. EPS has the
advantages of light weight, seismic, low cost, etc., Adding EPS can reduce soil
density, soil pressure, and settlement. Although the strength is reduced, adding
binders can compensate for the loss of strength, the strength of lightweight soil
can be adjusted according to engineering needs. The use of EPS is reasonable.

6- There isn't a comparison with the original soil with no additive, it's likely that
the soil with no admixture would exhibit better properties. A comparison with the
plain soil must be added.

Response: Thanks so much for your suggestion. based on your suggestion, we have added
Fig. 2 about the original soil with no additive on line 181.

7- what is it meant by curing agent? Slag and cement are most commonly used as
binding agents.

Response: Thanks so much for your comment. English expression error, replace " curing
agent " with " binding agents ". 

8-EPS could may cause deformation and settlement within time and loading, how the
authors justify the usage of it?

Response: Thank you for your careful review and valuable comments. Researches have
conducted relevant research on EPS lightweight soil, shown that the addition of EPS
to soil can reduce soil's density and weight, but the addition of binders, EPS
lightweight soil has a certain strength and deformation resistance. Due to EPS
lightweight characteristics, it reduces soil load and pressure, instead reduces
deformation and settlement. 

9- Is there any environmental effects of using EPS? slag within the soil? It most be
considered.

Response: Thank you for your careful review and valuable comments. In 34-35, 68-72, I
have modified and supplemented the above formulation and other relevant places. EPS
molecular structure is stable, does not release harmful substances and cause
chemical reactions to the soil. Researches analyzed the composition and properties
of slag, and studied the application of slag in the fields of building materials,
sewage treatment, agriculture and resource utilization. The results show that making
full use of the secondary resources of slag can improve the protection of the
environment, realize the harmless treatment of slag, and further improve the
residual value of slag.

Thank you again for your positive and constructive comments and suggestions on our
manuscript. We tried our best to improve the manuscript and made some revisions in
the manuscript. We hope you can accept our revised manuscript.

to Reviewers.docx
---

## [Editor Report · Decision Letter 1]

4 Jan 2024

Research on the Mechanical Properties of EPS Lightweight Soil Mixed with Slag

PONE-D-23-31063R1

Dear Dr. Xiang,

We’re pleased to inform you that your manuscript has been judged scientifically
suitable for publication and will be formally accepted for publication once it meets
all outstanding technical requirements.

Kind regards,

Abdullah Ekinci, PhD

Academic Editor

PLOS ONE
---

## [Editor Report · Acceptance letter]

11 Jan 2024

PONE-D-23-31063R1 

PLOS ONE

Dear Dr. Xiang, 

I'm pleased to inform you that your manuscript has been deemed suitable for
publication in PLOS ONE. Congratulations! Your manuscript is now being handed over
to our production team.

Kind regards, 

on behalf of

Assoc. Prof. Dr. Abdullah Ekinci 

Academic Editor

PLOS ONE